# Joint Developmental Trajectories of Perinatal Depression and Anxiety and Their Predictors: A Longitudinal Study

**DOI:** 10.3390/healthcare13111251

**Published:** 2025-05-26

**Authors:** Minhui Jiang, Han Zheng, Zhaohua Bao, Zhenhong Wu, Xiaomin Zheng, Yaling Feng

**Affiliations:** 1Department of Psychology, Affiliated Women’s Hospital of Jiangnan University, Wuxi 214002, China; a13961831712@163.com (M.J.); shinebao@sina.com (Z.B.); 2School of Medicine, Jiangnan University, Wuxi 214122, China; 3Department of Public Health, Wuxi Maternal and Child Health Hospital, Wuxi 214002, China; zh2971804096@163.com (H.Z.); zhenhongwu123@sina.com (Z.W.); 4Research Institute for Reproductive Health and Genetic Diseases, Wuxi Maternal and Child Health Hospital, Wuxi 214002, China

**Keywords:** perinatal, depression, anxiety, joint developmental trajectories, predictors

## Abstract

**Background:** Perinatal depression and anxiety can be experienced simultaneously and change over time. This study aimed to explore the independent and joint developmental trajectories and predictors of perinatal depression and anxiety. **Methods:** From January 2022 to December 2023, a total of 1062 pregnant women from Affiliated Women’s Hospital of Jiangnan University were surveyed for depression and anxiety symptoms using the Patient Health Questionnaire (PHQ-9) and Generalized Anxiety Disorder Scale (GAD-7) in early pregnancy (T1, 0–13^+6^ weeks), mid-term pregnancy (T2, 14–27^+6^ weeks), late pregnancy (T3, 28–41 weeks), and 42 days postpartum (T4). Parallel-process latent class growth model (PPLCGM) was performed to identify the joint developmental trajectories of perinatal depression and anxiety, and logistic regression was used to analyze factors of joint trajectories. **Results:** Perinatal depression and anxiety each showed four heterogeneous developmental trajectories, and three joint developmental trajectories were identified: “high–slightly decreasing depression and high decreasing anxiety group” (3%), “low–stable depression and low–stable anxiety group” (71%), and “moderate–slightly increasing depression and moderate–decreasing anxiety group” (26%). Adverse maternal history, history of anxiety and depression, and work stress were risk factors for the joint developmental trajectory of perinatal depression and anxiety, while regular exercise, paid work and social support were protective factors. **Conclusions:** Three joint developmental trajectories for perinatal depression and anxiety were identified, demonstrating group heterogeneity. Perinatal healthcare providers should pay attention to the mental health history of pregnant women, conduct multiple assessments of perinatal anxiety and depression, prioritize individuals with risk factors, and advocate for regular exercise, work participation, and provide greater social support.

## 1. Introduction

Perinatal depression and anxiety are very common globally, with prevalence rates as high as 5–30% [1,2,3]. If perinatal depression and anxiety are not treated promptly, they not only affect one’s physical and mental health and lead to adverse pregnancy outcomes, including spontaneous abortion, pre-eclampsia, cesarean section, preterm birth, and low birth weight [4,5,6], but also have long-term effects on the cognitive and emotional development of offspring, as well as contribute to behavioral problems and interpersonal relationship difficulties later in life [7,8]. In addition, perinatal mood disorders can decrease the rate of breastfeeding [9], and disrupt the quality of mother-infant attachment [10]. Given the high risk and prevalence of perinatal depression and anxiety, it is necessary to engage in extensive theoretical discussions and empirical research on the association between perinatal depression and anxiety.

Numerous studies have shown that perinatal depression and anxiety symptoms are heterogeneous, with a high degree of diversity in their onset, course, duration and severity [11,12,13]. Both domestic and international studies have shown that the prevalence of depression and anxiety varies at different stages of the perinatal period [14,15], and there is no definitive pattern regarding which has a higher or lower prevalence of mood disorders during pregnancy and postpartum. Two foreign studies on depression trajectories in perinatal women both found five trajectories, including trajectory categories of no depressive symptoms, depression during pregnancy, and postpartum depression [16,17]. Two domestic studies on pregnant women both identified three depression trajectories, including high symptom group, moderate symptom group and low symptom group [18,19]. Empirical studies of anxiety trajectories during the perinatal period are relatively limited compared to perinatal depression. A longitudinal study of perinatal anxiety among African women identified four distinct anxiety trajectory categories: low anxiety, increasing anxiety before and after childbirth, overall increasing anxiety, and transient high anxiety in the postpartum period [20]. Another study of potential trajectories of perinatal anxiety symptoms from pregnancy to the early postpartum period determined three trajectory groups: very low–stable, low–stable, and moderate–stable [21]. While these studies all indicate the existence of different categories of depression and anxiety trajectories during the perinatal period, providing evidence for longitudinal trajectory studies on perinatal depression and anxiety, existing studies are inconsistent in the number of trajectories, symptom continuity or variability, and vary in results depending on the study population, location and the duration of follow-up.

In previous individual-centered research, it can be seen that the independent developmental trajectories of antenatal depression and anxiety are very similar in number and shape [20,22]. Depression and anxiety in most pregnant women can be maintained at relatively low levels over time, while a small number of individuals show stable high levels of depression and anxiety or an increase after childbirth [17,20,22]. Furthermore, variable-centered studies have confirmed that depression and anxiety symptoms are significantly correlated and co-morbid [23,24]. Regarding the interaction between depression and anxiety at different stages of the perinatal period, many studies have found that prenatal anxiety and depressive symptoms predicted postpartum anxiety and depression [25,26,27]. So, do these results imply that there are common trends between perinatal depression and anxiety? Traditional variable-centered studies ignored the heterogeneity of developmental patterns of perinatal depression and anxiety, and it is difficult to determine the exact pattern of the relationship between perinatal depression and anxiety by examining the characteristics of perinatal depression and anxiety only at the level of the variable, without distinguishing the heterogeneity in the developmental patterns of these two. The strength of the individual-centered approach lies in identifying heterogeneous developmental trajectories of different types of perinatal depression and anxiety. Previous research has separately examined the heterogeneous developmental trajectories of perinatal depression and anxiety but has not simultaneously investigated the co-occurrence patterns of these two symptoms. Given the limitations of prior research, we employed an individual-centered approach (parallel process latent class growth modeling (PP-LCGM)) to explore the joint developmental trajectories of perinatal depression and anxiety. This approach allows for the identification of distinct developmental clusters based on the intra-individual joint trajectories of the two symptoms [28]. This method has been successfully applied in studies of the joint trajectories of loneliness, depressive symptoms, and social anxiety from childhood to adolescence [29], anxiety–depressive trait and trait aggression in adolescents [30], and depressive and anxiety symptoms in college students [31], but has not yet been used to explore the joint developmental trajectories of perinatal depression and anxiety. This study further explores the joint developmental trajectories of the two, thereby elucidating the probable reasons for the high correlation and comorbidity between perinatal depression and anxiety at the individual level.

The high correlation and co-morbidity between perinatal depression and anxiety implies that there may be common developmental trajectories and pathogenic factors for both, and the multiple trajectories of perinatal depression and anxiety also suggest that there may be specific risk factors leading to distinct symptom patterns. If high risk groups for perinatal depression and anxiety can be identified, as well as potential risk and protective factors, early monitoring, psychological health education, and cognitive behavioral therapy can be conducted to reduce the risk of severe depression and adverse perinatal outcomes [32]. Previous studies have indicated that a history of mental illness, pregnancy loss, unintended pregnancy, pregnancy complications, smoking, domestic violence, abuse history, life stress, and lack of social or partner support are risk factors for perinatal depression and anxiety [33,34,35,36,37]. Pregnancy complications, history of mental illness, and perinatal anxiety are associated with the high depression trajectory [12,37,38,39]. Low income, higher levels of stress, history of depression and lack of partner support are associated with the high anxiety trajectory group [13,20]. Although there have been several studies on the predictors of perinatal depression and anxiety, the research on the longitudinal joint trajectory of perinatal depression and anxiety and its related factors is still lacking. Notably, although previous studies have emphasized social support as a protective factor for perinatal depression and anxiety, may independently influence the developmental trends of depression and anxiety [33,35,40,41], there are currently no studies examining the effects of social support on the joint developmental trajectories of perinatal depression and anxiety.

Therefore, the present study aimed to examine the heterogeneous joint trajectories of perinatal depression and anxiety and to assess relevant predictive factors. The protective effect of social support is highlighted, providing empirical evidence for targeted early intervention and treatment.

## 2. Materials and Methods

### 2.1. Participants

This is a longitudinal study of perinatal depression and anxiety from the Affiliated Women’s Hospital of Jiangnan University. The study was carried out from January 2022 to December 2023, with a total of 1658 women selected from the outpatient department. Among them, 1062 women met the inclusion criteria of this study and were analyzed in this paper, while 596 women were lost to follow-up. The process of participant selection is presented in Appendix A. The attrition analysis showed that there were no statistically significant differences in age (t = −1.624, *p* = 0.104), education level (χ^2^ = 3.963, *p* = 0.138), and monthly income level (χ^2^ = 4.51, *p* = 0.105) between the participants who continued in the study and those who were lost to follow-up, indicating that the attrition of participants in this study was random. The survey questionnaires were completed anonymously and coded digitally. Participants were also informed that they could withdraw from the study at any time. This study obtained written consent and ethical approval from the Ethics Committee of the Affiliated Women’s Hospital of Jiangnan University (2023-01-0628-15).

### 2.2. Procedure

Research data were collected using the patient health questionnaire (PHQ-9) and generalized anxiety disorder scale (GAD-7) in the following four periods: early pregnancy (T1, 0–13^+6^ weeks); mid-pregnancy (T2, 14–27^+6^ weeks); late pregnancy (T3, 28–41 weeks); and postpartum 42 days (T4). Participants also completed the general information questionnaire and perceived social support scale (PSSS) at T1. The inclusion criteria for this study were as follows: (1) aged 18–40; (2) early pregnancy (before 13^+6^ weeks); and (3) voluntary informed consent. The exclusion criteria were as follows: (1) family history of mental illness; (2) severe heart disease, infectious disease, severe preeclampsia; and (3) withdrawal of informed consent, lack of cooperation, or incomplete questionnaires.

### 2.3. Research Tools

#### 2.3.1. General Information Questionnaire

The general information questionnaire included the following demographic data: age; monthly income level (<5000 CNY, 5000–10,000 CNY, >10,000 CNY); education level (college and below, undergraduate, master and above); planned pregnancy (yes, no); regular exercise (walking >5000 steps/day, no); paid work (yes, no); work stress (yes, no); adverse maternal history (yes, no); number of births (0, ≥1); gestational diabetes (yes, no); gestational hypertension (yes, no); history of anxiety (yes, no); history of depression (yes, no); preterm birth (yes, no); newborn sex (male, female); and delivery mode (cesarean section, vaginal delivery).

#### 2.3.2. Patient Health Questionnaire

The patient health questionnaire (PHQ-9) developed by Kroenke et al. [42] was used to assess the level of perinatal depression with a total of 9 items. The scale utilizes a four-point rating (0 = not at all, 3 = nearly every day). Scores range from 0 to 27. A higher PHQ-9 score indicates a higher severity of depression, with cutoff points of 5 and 10 signifying mild and moderate depression symptoms, respectively [42]. This measure showed relatively high internal consistency at each time point (Cronbach’s α: α_T1_ = 0.830, α_T2_ = 0.835, α_T3_ = 0.837, α_T4_ = 0.847).

#### 2.3.3. Generalized Anxiety Disorder Scale

The generalized anxiety disorder scale (GAD-7), developed by Spitzer et al. [43], was used to assess the level of perinatal anxiety. The scale consists of 7 items rated on a four-point scale (0 = not at all, 3 = nearly every day), with scores ranging from 0 to 21. Higher GAD-7 scores indicate more severe anxiety levels, with a cutoff point of 7 indicating the presence of anxiety symptoms [44]. This measure showed good internal consistency at each time point (Cronbach’s α: α_T1_ = 0.833, α_T2_ = 0.818, α_T3_ = 0.824, α_T4_ = 0.811).

#### 2.3.4. Perceived Social Support Scale

The perceived social support scale (PSSS) developed by Zimet et al. [45], translated and revised by Qianjin Jiang [46] was used to measure the perceived social support of the perinatal women. It consists of 12 items in 3 subscales, rated on a 7-point scale (1 = strongly disagree, 7 = strongly agree), with scores ranging from 12 to 84. The scale includes three dimensions: family support, friend support, and other support, with higher total scores indicating greater social support for the individual. The Cronbach’s coefficient α of the total scale in this study was 0.887, and the internal consistency (Cronbach’s αs) of family support, friend support, and other support in the subscales were 0.745, 0.712, and 0.747, respectively, which have reached the psychometric standard.

### 2.4. Data Analysis

Firstly, descriptive statistics were conducted on the research variables to explore the correlation between the levels of depression and anxiety at various measurement time points and their correlation with social support variables.

Secondly, the latent growth model and latent class growth model were constructed to examine the developmental trajectories and classes of perinatal depression and anxiety [47,48]. The latent growth model was used to investigate the trajectory of perinatal depression and anxiety changes, and whether there were significant individual differences in the initial level and development rate. The model was considered well-fitted when the confirmatory fit index (CFI) and the Tucker–Lewis Index (TLI) were ≥0.95, and the root mean square error of approximation (RMSEA) was <0.08 [49].

Subsequently, the latent class growth model (LCGM) was separately constructed for perinatal depression and anxiety to explore their potential categories. The following parameters were used to determine the optimal number of categories and the fit of the model: Akaike information criterion (AIC) [50]; Bayesian information criterion (BIC) [51]; a-BIC (smaller values indicate a better fit of the model with increasing class numbers) [52]; Entropy (entropy value above 0.70 suggests high classification accuracy) [53]; BLRT (boot-strapped likelihood ratio test); and VLMR (Vuong-lo-mendell-rubin likelihood ratio test). The acceptance of K group classification and rejection of k-1 group classification was based on results from BLRT and VLMR test reaching significance (*p* < 0.05) [54,55]; the proportion of each subgroup group was not less than 3% [56]. In addition to these fit indices, the practical interpretability of the trajectory classes should also be considered [55].

Furthermore, the parallel-process latent class growth model (PPLCGM) [48] was established to investigate the joint developmental trajectories of perinatal depression and anxiety. This model extends the typical univariate latent class growth model to parallel processes, considering multiple growth trajectories simultaneously [57].

Finally, multinomial logistic regression was developed to explore whether demographic variables and social support significantly predicted the joint developmental trajectories of perinatal depression and anxiety. Prior to the multinomial logistic regression analysis, we assessed the model for multicollinearity. The tolerance values for each independent variable were all well above 0.1, and the variance inflation factors (VIFs) were all below 10, indicating the absence of multicollinearity. This study used SPSS 23.0 for descriptive, correlation, and regression analysis, Mplus 8.3 for latent class growth model analysis, and full information maximum likelihood (FIML) analyses to handle missing values, minimizing biases in regression coefficient and standard error estimates [58].

## 3. Results

### 3.1. Descriptive Statistical Analysis and Correlation Analysis

A total of 1062 pregnant women completed 4 screenings for anxiety and depression from early pregnancy to 42 days postpartum. Descriptive statistics of participants’ demographic characteristics were shown in Table 1. Four measurements of perinatal depression and anxiety were significantly positively correlated and significantly negatively correlated with all variables of social support (Appendix A).

### 3.2. Latent Class Growth Analysis for Perinatal Depression and Anxiety

First, from the fit indices of the three models (Appendix A), it was found that compared with the linear and quadratic model, the free estimated latent growth model of perinatal depression and anxiety fitted relatively well. In addition to the non-significant rate of change for perinatal depression (σ^2^dep = 0.065, *p* = 0.092), the variances in the initial levels of perinatal depression and anxiety (σ^2^dep = 12.671, *p* < 0.001, σ^2^anx = 8.545, *p* < 0.001) and in the slope of perinatal anxiety (σ^2^anx = −0.607, *p* < 0.01) were statistically significant, indicating that there may be multiple subgroups with different trajectories of perinatal depression and anxiety symptoms, respectively, which laid the foundation for the latent class growth model analysis.

Next, the latent class growth model (LCGM) was used to identify the optimal number of five latent classes for the developmental trajectories of perinatal depression and anxiety, respectively (Table 2). For depression, a subgroup included in the five-class model accounted for only two and a half percent of the sample, leading to the exclusion of the five-class model. The AIC, BIC, and a-BIC values decreased gradually with an increase in trajectory numbers, and the decrease slowed down when it dropped to class four. For anxiety, similar patterns were observed with AIC, BIC, and a-BIC values showing a gradual decrease with an increase in trajectory numbers, and then the decrease was not significant when it dropped to class four. The results of the VLMR test indicated no significance in the five-class model, leading to its exclusion. Based on the model’s fit indices, both three-class and four-class models of perinatal depression and anxiety trajectories are acceptable. Consequently, trajectory plots were generated for both three-class and four-class models, respectively (Figure 1 and Appendix A). The four-class model of perinatal depression and anxiety trajectories, in comparison, allows for the identification of more refined subgroups, aligning with the detailed scoring criteria of the PHQ-9 and GAD-7. In summary, four-class model was chosen for the developmental trajectories of perinatal depression and anxiety.

Specifically, the trajectories of perinatal depression were classified into the following four classes: class 1 (n = 186) showed an overall moderate level, with an increasing trend in late pregnancy and postpartum, classified as the “moderate risk depression group”; class 2 (n = 204) presented an overall low level, with a slight increase in the postpartum period, classified as the “low risk depression group”; class 3 (n = 596) consistently maintained at a very low level, as the “consistently low depression group”; and class 4 (n = 76) showed an overall high-risk level, as the “high risk depression group” (Figure 1a). Similarly, the trajectories of perinatal anxiety were divided into the following 4 classes: class 1 (n = 539) consistently remained at a very low level, defined as the “consistently low anxiety group”; class 2 (n = 318) displayed a stable low level, as the “low risk anxiety group”; class 3 (n = 50) presented an overall high-risk level, with a peak in the late-pregnancy period, identified as the “high risk anxiety group”; and class 4 (n = 155) maintained a stable moderate level, hovering around the cut-off value of 7 points, classified as the “moderate risk anxiety group” (Figure 1b). The parameters of the interception and slope were shown in Appendix A.

### 3.3. Parallel-Process Latent Class Growth Analysis of Perinatal Depression and Anxiety

The parallel process latent growth analysis (Appendix A) indicated that the free estimated latent growth model had a better fit compared to linear and quadratic growth models (χ^2^(df) = 190.604(18), *p* < 0.001, RMSEA = 0.095, CFI = 0.983, TLI = 0.974, SRMR = 0.020). In addition to the non-significant rate of change for perinatal depression (σ^2^dep = 0.054, *p* = 0.146), the variances in the initial levels of perinatal depression and anxiety (σ^2^dep = 12.609, *p* < 0.001, σ^2^anx = 8.574, *p* < 0.001) and in the slope of anxiety (σ^2^anx = −0.586, *p* = 0.02) were statistically significant, indicating the presence of multiple groups with different trajectories of perinatal depression and anxiety symptoms, and that latent class analysis using parallel processes for both was necessary.

Then, the parallel process latent class growth model was developed for perinatal depression and anxiety, extracting from one to five latent classes to identify the optimal number of the joint developmental trajectories of perinatal depression and anxiety (Table 3). As the number of trajectories increased, the values of AIC, BIC, and a-BIC decreased gradually, and the decrease slowed down when reaching class three. The values of VLMR and BLRT suggested that the models of class two and three were acceptable. Based on the fit indices, both two- and three-class models are acceptable. Consequently, we generated trajectory plots for two- and three-class models, respectively (Figure 2 and Appendix A). In comparison, the two-class model overlooks a high-risk group. Taking all into account, the three-class model was determined as the best-fitting model for joint developmental trajectories of perinatal depression and anxiety.

Based on this model, the joint developmental trajectories of perinatal depression and anxiety were classified into the following 3 classes (Figure 2): class 1 was the smallest, approximately 3% (n = 33), with high levels of perinatal depression and anxiety, and a significant decrease in perinatal anxiety, which was named “high–slightly decreasing depression and high decreasing anxiety group” (Intercept: I_dep_ = 14.415, *p* < 0.001, I_anx_ = 9.961, *p* < 0.001, Slope: S_dep_ = −0.291, *p* = 0.402, S_anx_ = −1.339, *p* < 0.001); class 2 consisted of 71% of pregnant women (n = 754) with consistently low levels of perinatal depression and anxiety, named “low–stable depression and low–stable anxiety group” (I_dep_ = 1.293, *p* < 0.001, I_anx_ = 1.744, *p* < 0.001, S_dep_ = 0.029, *p* = 0.344, S_anx_ = −0.135, *p* = 0.03); class 3 included 26% of pregnant women (n = 275) with moderate initial levels of depression and anxiety, and a decreasing trend in anxiety and a slightly increasing trend in depression, named “moderate–slightly increasing depression and moderate–decreasing anxiety group” (I_dep_ = 6.925, *p* < 0.001, I_anx_ = 5.804, *p* < 0.001, S_dep_ = 0.144, *p* = 0.375, S_anx_ = −0.769, *p* < 0.001). The levels of perinatal depression and anxiety for the three groups at 4 measurement points were shown in Appendix A.

### 3.4. Predictors of Joint Developmental Trajectories of Perinatal Depression and Anxiety

Using demographic variables, perinatal-related information and social support at baseline as independent variables, the classes of the joint trajectory of perinatal depression and anxiety as dependent variables, and the low–stable depression and low–stable anxiety group as the reference group, multinomial logistic regression analysis were used to examine the predictors of joint developmental trajectories of perinatal depression and anxiety (Table 4). The results found that pregnant women with adverse maternal history, history of anxiety and depression were 4.875 times (95% CI: 1.260–18.857), 10.069 times (95% CI: 1.289–78.679), and 9.515 times (95% CI: 1.437–63.007) more likely to belong to the high–slightly decreasing depression and high decreasing anxiety group, respectively. Pregnant women with job stress, history of previous anxiety and depression were 5.251, 12.165 and 4.127 times more likely to belong to the moderate–slightly increasing depression and moderate–decreasing anxiety group, respectively. However, pregnant women with regular exercise (OR: 0.533) and paid work (OR: 0.369) were less likely to belong to the moderate–slightly increasing depression and moderate–decreasing anxiety group. Additionally, higher levels of social support reduced the odds of being allocated in the high–slightly decreasing depression and high decreasing anxiety group and the moderate–slightly increasing depression and moderate–decreasing anxiety group (ORs: 0.556–0.754).

## 4. Discussion

### 4.1. Characteristics of Independent Developmental Trajectories of Perinatal Depression and Anxiety

This study identified four perinatal depression trajectory groups and four perinatal anxiety trajectory groups. The four depression trajectory groups were: moderate-risk group, low-risk group, consistently low group and high-risk group. Around 75% of pregnant women belonged to the low-risk group and consistently low group, with depression scores below the clinically significant threshold. This finding was generally consistent with those of prior studies employing diverse measurement and statistical methodologies across varying environmental contexts [12,18,22,59,60]. The high risk group was characterized by a persistent high risk level of depression, indicating that for the majority of women suffering from postpartum depression, depressive symptoms may have appeared even before pregnancy, during pregnancy, adolescence, or in adulthood, representing a continuation and variation in early mental health problems [61,62]. In our study, the high risk group for depression showed a significantly increasing trend of depression scores in the postpartum period, while a study conducted in Norway [63] found a decreasing trend only in postpartum depression trajectories. This may be related to the development of the country, higher level of education and the relative superiority of social resources, which could potentially reduce the risk of depressive symptoms. A longitudinal study conducted in China identified groups with prenatal and postpartum depression [19], with proportions aligning with the high-risk group in this study. This suggests that the risk of perinatal depression may manifest at different stages, both prenatally and postnatally. Similarly, the four perinatal anxiety trajectory groups also showed consistently low group, low-risk group, high-risk group, and moderate-risk group, which is both similar and specific to previous studies. Our findings revealed that over 80% of women experienced either very low or low levels of anxiety symptoms throughout the entire period, which aligns with the research on anxiety trajectories of 1445 perinatal women in Canada [64] and 778 women in West Africa [20]. Less than 1/5 of pregnant women exhibited mild to moderate anxiety symptoms, and the trends of the 4 trajectories were similar from early pregnancy to 42 days postpartum, with a declining trend in anxiety levels at 42 days postpartum. In contrast, Barthel et al. [20] found less than one-fifth of pregnant women displayed three different moderate to high anxiety trajectory groups. However, both studies revealed a subgroup of women whose anxiety scores increased around delivery and subsequently decreased, indicating the prevalence of delivery-related anxiety and the necessity for timely psychological interventions before and after delivery. Meanwhile, perinatal anxiety and depressive symptoms show a degree of similarity in trends, supporting the idea that depression and anxiety are independent and interdependent.

### 4.2. Characteristics of Joint Developmental Trajectories of Perinatal Depression and Anxiety

The present study identified three joint developmental trajectories of perinatal depression and anxiety, among the three groups, the “high–slightly decreasing depression and high decreasing anxiety group” had the smallest proportion of pregnant women, with both depression and anxiety levels remaining high, and the “low–stable depression and low–stable anxiety group” had the highest proportion of pregnant women, with consistently low levels of depression and anxiety, while the “moderate–slightly increasing depression and moderate–decreasing anxiety group” had a moderate proportion of pregnant women, with moderate initial levels of depression and anxiety, followed by a declining trend in anxiety. This finding indicated that the majority of pregnant women belonged to the low–stable depression and low–stable anxiety group, suggesting that for most pregnant women, depressive and anxiety symptoms were generally low and stable, consistent with other research [12,59,60,64,65]. Only a minority of pregnant women belonged to the high–slightly decreasing depression and high decreasing anxiety group, indicating that the prevalence of co-morbid high-risk depression and anxiety among pregnant women is not high, which may also be related to our selection of individuals with fewer emotional symptoms as the study participants. The results of the joint developmental trajectories revealed a certain degree of similarity and commonality in the initial levels and trends of anxiety and depression symptoms among the three groups, supporting their comorbidity [66]. This implied that regular perinatal screening should not only focus on depressive mood but also be attentive to all emotional disorders, including anxiety. Additionally, it was interesting to note that the trajectories of perinatal anxiety symptoms exhibited varying degrees of decreasing trends in all three groups, especially more pronounced in the postpartum period, further confirming the findings of Buist et al. [67]. Whereas the trajectory of perinatal depression still had the risk of increasing, indicating that pregnant women need to possess emotional regulation strategies and problem-solving skills to effectively cope with their distress and prevent postpartum negative emotions [68].

### 4.3. Predictors of Joint Developmental Trajectories of Perinatal Depression and Anxiety

This study identified risk and protective factors associated with the joint developmental trajectories of perinatal depression and anxiety. Pregnant women with a history of adverse pregnancy outcomes, anxiety, and depression were more likely to belong to the high–slightly decreasing depression and high decreasing anxiety group, and those with high work stress, history of anxiety and depression were more likely to belong to the moderate–slightly increasing depression and moderate–decreasing anxiety group. It can be seen that adverse maternal history is an important factor influencing perinatal depression and anxiety [36]. Women who have experienced adverse pregnancy outcomes often worry early in pregnancy, fearing the recurrence of miscarriage, fetal deformities, and preterm birth. Persistent anxiety may diminish or disappear after the successful delivery. The history of previous anxiety and depression is major risk factor, and several studies have confirmed that women who have experienced anxiety and depression in the past are more likely to be depressed and anxious during pregnancy and postpartum [12,33,34,69,70]. Indeed, for individuals with a history of anxiety and depression, pregnancy and childbirth as stressful events can intensify stress responses, leading to increased emotional instability and vulnerability. Work stress, as one of the factors affecting maternal mental health, has been mentioned in previous findings [12,64]. The dual stress of work and childbirth not only triggers hormonal changes, such as, activation of the HPA axis, the release of corticotropin releasing hormone (CRH), and cortisol levels, but may also exacerbate the physical discomforts associated with pregnancy and increase susceptibility and vulnerability to perinatal depression or anxiety [71].

Furthermore, pregnant women who engage in regular exercise and paid work are not categorized into the moderate–slightly increasing depression and moderate–decreasing anxiety group, indicating that exercise and paid work seem to be protective factors. This finding, though interesting, is not difficult to understand, studies have shown that physical exercise during pregnancy can reduce the incidence and severity of perinatal depression [72,73]. It can be seen that appropriate exercise can enhance physical fitness and is a beneficial remedy for the smooth delivery and emotional regulation of pregnant women. Similarly, a previous study has confirmed that the mental health and quality of life scores of mothers with paid work were significantly higher than those of mothers who did not work [74]. Therefore, paid work with the appropriate intensity can provide some economic security and social support, reflect personal value, and reduce inexplicable anxiety in pregnant women.

Additionally, it has been found that social support significantly increases the likelihood of individuals belonging to the low–stable depression and low–stable anxiety group, indicating that social support as a protective factor, can significantly reduce the risk of perinatal depression and anxiety [20,75]. According to the stress-buffering model of social support, social support acts as a buffer between perceived stress and mental health, it can mitigate the impact of stress on mental health by alleviating individual stress appraisal responses [76]. Given that pregnancy and childbirth are stressful events, adequate social support is particularly important for pregnant women to resist stress and accumulate positive emotions. Hormonal fluctuations during pregnancy, including variations in oxytocin, cortisol, and serotonin, are closely associated with mood regulation. The release of oxytocin enhances maternal-infant bonding and alleviates stress [10]. Elevated cortisol levels during pregnancy may influence maternal mood [77], while social support buffers stress responses, mitigating excessive cortisol secretion [78]. The increase in oxytocin around childbirth and the sharp decline in estrogen explain the observed decrease in anxiety symptoms and the persistent risk of depression in this study, highlighting the importance of sustained social support in preventing depression. Therefore, at the family level, mental health education programs for partners and family members can be implemented, along with providing effective companionship to enhance their ability to recognize emotional changes in pregnant women, thereby helping to alleviate emotional distress and prevent depressive symptoms [79]. At the community and peer level, mutual support groups or online support communities for pregnant women can be established to promote peer interaction and mutual support [79], and regular mental health promotion activities can be organized to disseminate knowledge of perinatal mood management. From a healthcare policy perspective, promoting the inclusion of perinatal mental health screening in routine prenatal check-ups ensures that high-risk pregnant women receive timely interventions such as mindfulness training and cognitive behavioral therapy [80].

### 4.4. Strengths, Limitations, and Further Research

The current study displays several major strengths and conducted repeated assessments of anxiety and depression during early pregnancy, mid-pregnancy, late pregnancy, and postpartum from a dynamic longitudinal perspective. It analyzes the independent and joint developmental trajectories of perinatal depression and anxiety, providing important insights and reference value for clinical diagnosis and treatment. Additionally, the study explores the risk and protective factors of the joint developmental trajectories of perinatal depression and anxiety, offering clinical guidance for screening and prevention of perinatal mental health.

The study has several limitations. Firstly, the current sample size may not be sufficient for fine identification of joint trajectories of depression and anxiety. Previous studies with larger sample sizes have been able to differentiate five or more heterogeneous trajectories of anxiety and depression [64]. The relatively small size of the high symptom trajectory group may affect the precision of the correlation between predictors and each group. Future research could benefit from expanding the sample size to more precisely identify joint developmental trajectories and predictors. Secondly, this study only investigated the 4 following periods: early pregnancy; mid-pregnancy; late pregnancy; and 42 days postpartum. Future research could extend the investigation period to one year postpartum to comprehensively characterize the entire developmental trend of perinatal depression and anxiety. Thirdly, this study is limited by its inability to encompass all potential influencing factors, and the use of self-report questionnaires to assess psychosocial variables may introduce response bias. Future research could consider network analysis to accurately explore the association patterns between potential influencing factors and perinatal anxiety and depressive symptoms. Lastly, the study does not investigate which pregnant women received standard treatments for depression and anxiety, which could potentially alter their trajectories of perinatal mood disorders. Future research could explore whether standardized psychological interventions and necessary drug treatments could change the trajectories of perinatal anxiety and depression.

## 5. Conclusions

We found three joint developmental trajectories of perinatal anxiety and depression. Adverse maternal history, history of anxiety and depression, and work stress were risk factors, while regular exercise, paid work and social support served as protective factors. Perinatal healthcare providers should pay attention to the mental health history of pregnant women, conduct multiple assessments of perinatal anxiety and depression, prioritize individuals with risk factors, encourage pregnant women to engage in regular exercise, participate in work, and provide them with greater social support. This study offers practical guidance for early screening, dynamic monitoring, and personalized interventions for perinatal depression and anxiety.

## Figures and Tables

**Figure 1 healthcare-13-01251-f001:**
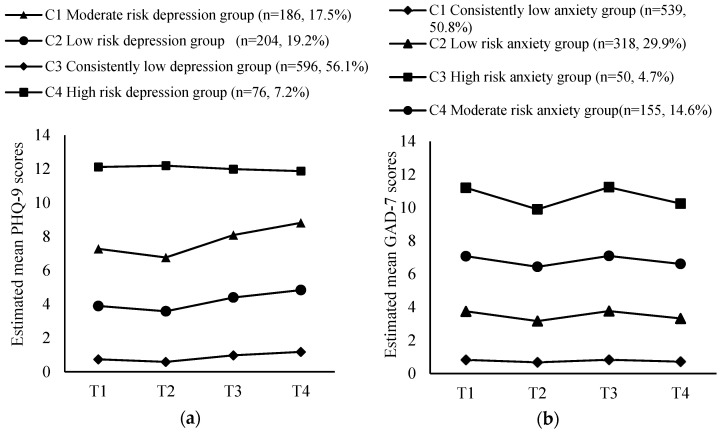
(**a**) Developmental trajectory of perinatal depression; (**b**) Developmental trajectory of perinatal anxiety.

**Figure 2 healthcare-13-01251-f002:**
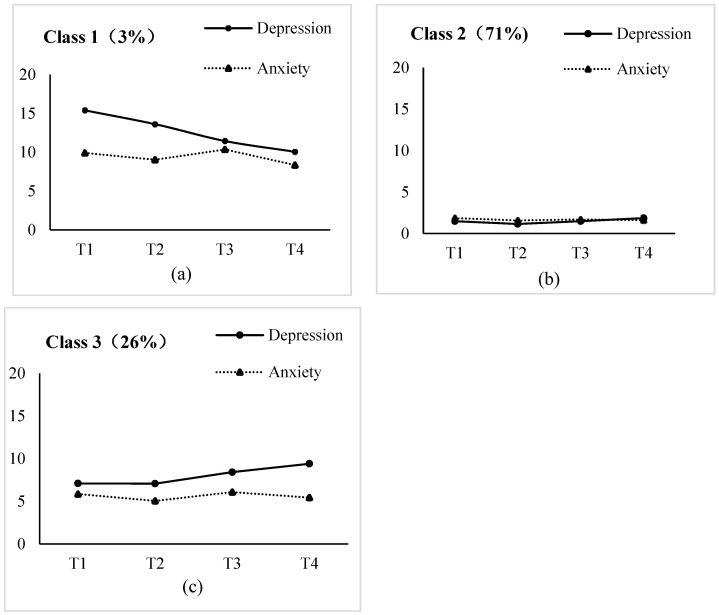
Joint developmental trajectory classes of perinatal depression and anxiety: (**a**) Class 1—high–slightly decreasing depression and high decreasing anxiety group; (**b**) Class 2—low–stable depression and low–stable anxiety group; (**c**) Class 3—moderate–slightly increasing depression and moderate–decreasing anxiety group.

**Table 1 healthcare-13-01251-t001:** Demographic characteristics of participants (n = 1062).

Variables	Categories	Mean ± SD/N (%)	Variables	Categories	Mean ± SD/N (%)
Age		29.203 ± 3.853	Number of births	0	607 (57.2)
Monthly income	<5000 CNY	211 (19.9)		≥1	455 (42.8)
	5000–10,000 CNY	562 (52.9)	Gestational diabetes	Yes	151 (14.2)
	>10,000 CNY	289 (27.2)		No	911 (85.8)
Education level	College and below	604 (56.9)	Gestational hypertension	Yes	133 (12.5)
	Undergraduate	405 (38.1)		No	929 (87.5)
	Master and above	53 (5.0)	History of anxiety	Yes	56 (5.3)
Planned pregnancy	Yes	671 (63.2)		No	1006 (94.7)
	No	391 (36.8)	History of depression	Yes	66 (6.2)
Regular exercise	Yes	303 (28.5)		No	996 (93.8)
	No	759 (71.5)	Preterm birth	Yes	92 (8.7)
Paid work	Yes	599 (56.4)		No	970 (91.3)
	No	463 (43.6)	Newborn sex	Male	537 (50.6)
Work stress	Yes	256 (24.1)		Female	525 (49.4)
	No	806 (75.9)	Delivery mode	Cesarean section	473 (44.5)
Adverse maternal history	Yes	381 (35.9)		Vaginal delivery	589 (55.5)
	No	681 (64.1)			

**Table 2 healthcare-13-01251-t002:** Model fit indices for latent class growth models of perinatal depression and anxiety.

Class	AIC	BIC	a-BIC	BLRT	VLMR	Entropy	Numbers of Each Class
Dep							
1	23,603.303	23,643.046	23,617.637				
2	20,138.057	20,192.704	20,157.766	<0.001	<0.001	0.954	751/311
3	18,837.218	18,906.769	18,862.303	<0.001	0.0005	0.961	679/111/272
4	18,335.062	18,419.517	18,365.522	<0.001	0.008	0.933	186/204/596/76
5	17,945.721	18,045.079	17,981.555	<0.001	0.0198	0.933	200/176/570/89/27
Anx							
1	21,733.744	21,773.487	21,748.078				
2	19,028.138	19,082.785	19,047.847	<0.001	<0.001	0.939	248/814
3	18,022.738	18,092.289	18,047.823	<0.001	0.0079	0.908	125/323/614
4	17,535.358	17,619.813	17,565.818	<0.001	0.0060	0.909	539/318/50/155
5	17,374.588	17,473.946	17,410.422	<0.001	0.1224	0.875	292/159/471/100/40

Note: Dep represents Depression, Anx represents Anxiety.

**Table 3 healthcare-13-01251-t003:** Fit indices of parallel process latent class growth model for perinatal depression and anxiety.

Number of Class	AIC	BIC	a-BIC	BLRT	VLMR	Entropy	Number of Each Class
1	35,497.061	35,591.451	35,531.104				
2	35,110.937	35,230.167	35,153.939	<0.001	0.0023	0.866	779/283
3	34,699.258	34,843.327	34,751.218	<0.001	<0.001	0.926	33/754/275
4	34,474.581	34,643.490	34,535.500	<0.001	0.1428	0.928	716/252/32/62
5	34,446.357	34,640.105	34,516.234	<0.001	0.3715	0.938	62/1/717/251/31

**Table 4 healthcare-13-01251-t004:** Logistic regression analysis of demographic and psychosocial factors on the subgroups of the joint developmental trajectories of perinatal depression and anxiety.

Predictor Variables	High–Slightly Decreasing Depression and High Decreasing Anxiety Group		Moderate–Slightly Increasing Depression and Moderate–Decreasing Anxiety Group	
	OR	95% CI	OR	95% CI
Age	1.070	0.921–1.244	0.974	0.918–1.033
Monthly income (>10,000 CNY as reference):				
<5000 CNY	5.658	0.641–49.967	0.717	0.350–1.472
5000–10,000 CNY	7.444	1.003–55.245	0.718	0.408–1.261
Educational level (Master and above as reference):				
College and below	0.257	0.010–6.554	0.899	0.266–3.032
Undergraduate	0.738	0.029–18.719	0.904	0.276–2.956
Planned pregnancy: yes vs. no	0.523	0.121–2.256	0.993	0.583–1.693
Regular exercise: yes vs. no	1.035	0.192–5.566	0.533 *	0.308–0.923
Paid work: yes vs. no	0.49	0.112–2.137	0.369 ***	0.214–0.638
Work stress: yes vs. no	2.211	0.665–7.347	5.251 ***	3.061–9.010
Adverse maternal history: yes vs. no	4.875 *	1.260–18.857	1.674	0.975–2.873
Number of births: 0 vs. ≥1	1.345	0.332–5.451	0.743	0.466–1.185
Gestational diabetes: yes vs. no	0.508	0.059–4.358	0.941	0.488–1.814
Gestational hypertension: yes vs. no	0.977	0.156–6.112	1.403	0.694–2.835
History of anxiety: yes vs. no	10.069 *	1.289–78.679	12.165 ***	3.470–42.645
History of depression: yes vs. no	9.515 *	1.437–63.007	4.127 *	1.340–12.708
Preterm birth ^a^: yes vs. no	1.078	0.088–13.213	1.083	0.488–2.407
Newborn sex ^a^: male vs. female	2.275	0.501–10.340	0.677	0.432–1.083
Delivery mode ^a^: cesarean section vs. vaginal delivery	0.421	0.128–1.390	0.98	0.599–1.605
Social support	0.556 ***	0.500–0.620	0.754 ***	0.724–0.786

Using the low–stable depression and low–stable anxiety group as reference groups, ^a^ represented data for these three variables were collected at T4. OR: odds ratio, CI: confidence interval. * *p* < 0.05, *** *p* < 0.001.

## Data Availability

The datasets used and/or analyzed during the current study are available from the corresponding author upon reasonable request.

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
