# Peer review of "Joint Developmental Trajectories of Perinatal Depression and Anxiety and Their Predictors: A Longitudinal Study"

_healthcare, 2025, doi:10.3390/healthcare13111251_

Round 1
Reviewer 1 Report
Comments and Suggestions for Authors
In the manuscript, Dr.Jiang and other co-authors conducted a joint analysis of perinatal depression and anxiety using parallel-process latent class growth model, aiming to explore the independent and joint developmental trajectories and predictors of perinatal depression and anxiety. This manuscript is well-organized and utilize advanced statistical methods to investigate underling joint subgroups of depression and anxiety. However, this manuscript lacks explanation in methods and results, as well as some small details which needs to be fixed. I have multiple major concerns are as follows:
- The introduction is in lack of citations for some sentences, please add citations for the following:
- Line 75-76: small number of individuals show stable high levels of depression and anxiety or an increase after childbirth.
- Line 88: as far as I know, there were lots of similar analyses being done in terms of joint developmental trajectory analysis. I would like authors to have a brief discussion with those analyses, along with necessary citations
- Data analysis part is too short and lack of essential information and citations of certain methods used in the analysis
- Though I acknowledged that it is not your focus to discuss the methods of model fitness in this paper, but I would like authors to include citations of some of model statistics used in the analysis , such as a-BIC, Entropy, BLRT and VLMR. It’s useful to include the citation of original papers of these statistics so that audiences who are interested in could refer to the paper for them.
- Multivariate logistic regression was mistakenly specified in this paper. I think you mean multinomial logistic regression. Multinomial logistic regression predicts a categorical outcome with more than two categories, while multivariate logistic regression predicts multiple binary outcomes simultaneously. Please change to multinomial logistic regression throughout the manuscript.
- Some tables and figures in Results and Supplemental files are in low quality and needs to be modified, and
- In Table 1, please give p-values alongside with mean +/- SD or indicate by * for which variables are significant by category groups
- Table S2 is not clear to me and need further clarification: 1) what is P? Is it overall model significance level or p-value for any covariates? 2) what does * and *** mean in the table, a footnote is needed. 3) in the growth model, did you adjust for other covariates? 4) what does free estimated model mean?
- In general, how did you select the number of groups for each latent class model? For the results of latent class growth model for anxiety (lines 238-245), 4-class model was selected, while only 50 subjects were in one of the groups. In my opinion, though model fit indices is one of the important considerations when selecting the model, you should also consider the trajectory and practical explanation from clinical angle. I would like the authors to justify why not using 3-class model for anxiety and why not merging this group with moderate anxiety groups (in Figure 3b).
- Figure 1 only had labels for the first two groups. Please check and include labels of all groups (with names, n and percentage).
- In Results section 3.3 lines 267-275, where are those results? I cannot find results you displayed in any tables, figures or in supplementary documents. Please add a table of these results.
- In lines 281-282, 3-class model was selected for the joint latent class growth model. I did agree with you to stay with 3-class model. But again, I want to emphasize that this decision should not be made by only looking at model fit indices solely. Model fit indices, trajectories, as well as judgement from clinical and practical angles, all of them should be considered when selecting the best model. I would like authors to include figure of trajectories of 2-class model in the supplemental files and briefly discuss why you selected 3-class model over 2-class model.
- In Table 4 results from multinomial logistic regression, it seems that you used all possible predictors in the model. Multicollinearity issue may occur while using so many predictors in one model. I would like authors to check multicollinearity issue for this model. Drop variables with high correlation and only include variables that without high correlation in the model.
Please go through the manuscript and revise as much as possible. Improving the quality of the exposition can make an interesting study appealing to more readers.
Reviewer 2 Report
Comments and Suggestions for Authors
The study addresses an essential and timely topic with significant implications for perinatal health and maternal well-being. The chosen approach provides valuable insights into the joint trajectories of depression and anxiety during this period, and the findings have the potential to support the development of monitoring and intervention strategies. However, some refinements could further enhance the clarity, coherence, and impact of the research. Below, we present suggestions that may contribute to improving the manuscript.
Results
• The presentation of the results is clear and well-structured, but organizing them in a more fluid manner could improve readability. Providing a brief summary of the identified trajectories (three joint profiles of depression and anxiety) before delving into the statistical findings would help contextualize the reader.
• The tables are informative and well-prepared, but including a schematic figure summarizing the key findings could facilitate understanding and make interpretation more intuitive.
• The statistical model is well described, but the criteria used to select the optimal number of classes (AIC, BIC, VLMR, BLRT) could be presented more concisely, emphasizing only the essential values for decision-making.
Discussion
• The discussion is well-constructed and well-grounded, but it could be further developed regarding the implications of the findings. Additional reflections could strengthen the impact of the research.
• The comparison with previous studies could be expanded, highlighting similarities and differences in the identified trajectory patterns. Some references are briefly mentioned, and it would be helpful to provide more detailed explanations of the reasons behind the observed discrepancies.
• The discussion of risk and protective factors is appropriate, but further elaboration on the impact of sample attrition and potential selection biases would be beneficial, given that a significant portion of the initial sample did not complete all study phases.
• The limitations section already addresses important aspects, such as sample size and the lack of data on psychological and pharmacological interventions. However, it could provide more direct suggestions for future research, such as a longer follow-up period postpartum (e.g., up to one year) and an evaluation of the impact of early interventions for high-risk pregnant women.
Conclusion
The conclusion effectively synthesizes the findings and reinforces the importance of the study. However, it could further emphasize the practical implications for healthcare professionals. Highlighting how these findings can assist in screening, monitoring, and developing strategies for pregnant women at higher risk of mental health disorders would make this section even more relevant.
• The role of social support networks as a protective factor is correctly highlighted but could be further explored. It is suggested to include more specific recommendations on interventions that could be implemented to reduce the incidence of depressive and anxiety symptoms during the perinatal period.
Final Considerations
The study makes a valuable contribution to the field, addressing a highly relevant topic in perinatal mental health. The suggestions provided aim to further enhance the manuscript’s clarity and impact, strengthening its practical applicability and reinforcing its connection to the existing literature. We commend the authors for their work and hope these considerations contribute to the improvement of the study.
Reviewer 3 Report
Comments and Suggestions for Authors
The author's study provides an in-depth exploration of the potential developmental trajectories of comorbid anxiety and depression in pregnant women, offering strong supporting evidence. This research is highly significant, as emotional disorders during pregnancy are quite common and can have potential adverse effects on a family. However, I have a few minor suggestions that might further enhance the quality of the paper.
In the introduction part, the author mentioned that several risk factors were related to depression and anxiety. However, smoking and domestic violence, despite being related to both, seemed to be not included in the study. The inability to include all potential risk factors is also a limitation of this study. Besides, there may be bias since the survey is based on self-reports from participants.
Reviewer 4 Report
Comments and Suggestions for Authors
The paper concerns an important issue of a woman's health and threats to the proper development of her child.
The authors focused their research on the goal of exploring (1) the independent and (2) joint developmental trajectories and (3) predictors of perinatal depression and anxiety.
Longitudinal studies were conducted, which require repeated measurements at several points in time. This type of research is considered difficult, but at the same time very valuable due to the information it allows to obtain about changes over time. These changes are often given the status of developmental changes. Recognizing the states/values/studied variables (here: the intensity of anxiety and depression) at several points in time also provides a basis for planning adequate therapy, applying forms of support adapted to the actual needs of the woman - taking into account her life situation and health condition, including medical history. This approach is to some extent close to the idea of ​​personalized medicine.
In the discussed study, four measurements were made: three stages of pregnancy (approximately every two to three months) and the postpartum stage (42 days after delivery). Data were collected using the Patient Health Questionnaire(PHQ-9), Generalized Anxiety Disorder Scale (GAD-7), The Perceived Social Support Scale (PSSS), and a survey collecting basic data, sociodemographic data. (I have a question: were the data on support also collected four times?)
The collected data were subjected to statistical analysis, which resulted in the identification of specific developmental trajectories/changes in the intensity of depressive symptoms and anxiety. Three developmental change trajectories were identified, concerning changes in depression and anxiety at the same time (joint developmental trajectories). The results of the analyses were presented graphically, which facilitates the perception of the content. The authors note the consistency of their results with the results of previous studies on depression and anxiety in pregnant women and the perinatal period, which are available in specialist literature. The studies referred to in the article resulted in the identification of three trajectories connecting the course of changes in depression and anxiety and their predictors, with particular emphasis on support, physical activity of the woman and her situation in the work environment, including the financial situation. The importance of physical activity, the situation in the work environment and support turned out to be significant and specifically related to changes in the intensity of symptoms of depression and anxiety. Their constellation may constitute both a protective factor and a risk factor for changes in depression and anxiety. In explaining the effects of support, it is worth referring more clearly to, among others: to hormonal changes during pregnancy and the positive effects of various forms of support, such as the “happiness hormones”, the level of which may vary in the subsequent trimesters of pregnancy.
The study took into account selected sociodemographic factors, the occurrence of symptoms of depression, anxiety, participation in their therapy, previous pregnancies and others. It would be interesting to collect data on tokophobia, which may result in experiencing anxiety during pregnancy.
The article is written in an interesting way and one can expect great interest in its content among various groups of readers.
Round 2
Reviewer 1 Report
Comments and Suggestions for Authors
The authors has successfully addressed all of my comments and concerns. Now the contents of manuscript has been significantly improved compared to the original version. The quality of presentation and interest to readers are high based on my judgement. I think now the manuscript meets all criteria for this journal. Hence, I would recommend to accepting this manuscript in the current form.
Author Response
Comments 1: The authors has successfully addressed all of my comments and concerns. Now the contents of manuscript has been significantly improved compared to the original version. The quality of presentation and interest to readers are high based on my judgement. I think now the manuscript meets all criteria for this journal. Hence, I would recommend to accepting this manuscript in the current form.
Response 1: We sincerely appreciate your careful review of our manuscript and the high evaluation you have given. Your recognition holds great significance for our entire research team and has significantly boosted our morale. During the revision process, we deeply experienced your professionalism and rigor, and your comments guided us in improving the quality of the manuscript.
In response to your valuable suggestions, we have conducted comprehensive and in-depth revisions. Each recommendation prompted us to re-examine the content of the manuscript, and we have made considerable efforts to refine research details, optimize logical structure, and enhance language expression. Through these revisions, the manuscript has shown marked improvement in all aspects, which would not have been possible without your expert guidance.
We sincerely hope that this manuscript will be successfully published and contribute to the research in this field. We also look forward to it attracting attention and discussion from more scholars. Should any further modifications or additional information be required during the publication process, we will fully cooperate and actively complete all necessary tasks.